# Towards automated recipe genre classification using semi-supervised learning

**Nazmus Sakib**[ID][1]☯*, **G. M. Shahariar**[2]☯, **Md. Mohsinul Kabir**[1]‡, **Md. Kamrul Hasan**[1]‡, **Hasan Mahmud**[1]☯*

**1** SSL Lab, Dept. of CSE, Islamic University of Technology, Dhaka, Bangladesh, **2** Dept. of CSE, Ahsanullah University of Science and Technology, Dhaka, Bangladesh

☯ These authors contributed equally to this work.
‡ MMK and MKH also contributed equally to this work.
* nazmussakib009@gmail.com (NS); hasank@iut-dhaka.edu (HM)

## Abstract

Sharing cooking recipes is a great way to exchange culinary ideas and provide instructions for food preparation. However, categorizing raw recipes found online into appropriate food genres can be challenging due to a lack of adequate labeled data. In this study, we present a dataset named the "Assorted, Archetypal, and Annotated Two Million Extended (3A2M+) Cooking Recipe Dataset" that contains two million culinary recipes labeled in respective categories with extended named entities extracted from recipe descriptions. This collection of data includes various features such as title, NER, directions, and extended NER, as well as nine different labels representing genres including bakery, drinks, non-veg, vegetables, fast food, cereals, meals, sides, and fusions. The proposed pipeline named 3A2M+ extends the size of the Named Entity Recognition (NER) list to address missing named entities like heat, time or process from the recipe directions using two NER extraction tools. 3A2M+ dataset provides a comprehensive solution to the various challenging recipe-related tasks, including classification, named entity recognition, and recipe generation. Furthermore, we have demonstrated traditional machine learning, deep learning and pre-trained language models to classify the recipes into their corresponding genre and achieved an overall accuracy of 98.6%. Our investigation indicates that the title feature played a more significant role in classifying the genre.

**Data Availability Statement:** All relevant data for this study are publicly available from the Kaggle repositories 3A2M Dataset (https://www.kaggle.com/datasets/nazmussakibrupol/3a2m-cooking-recipe-dataset), 3A2M+ Dataset (https://www.

## 1 Introduction

Food recipe classification is a crucial aspect of understanding and organizing recipe data, particularly in the field of machine learning and deep learning. With the growing availability of large food recipe datasets (RecipeNLG [1] and Recipe1M+ [2]) on the internet, researchers have been able to train and test deep learning models in the culinary domain [3, 4]. Recipe data includes a wealth of information, including ingredients, cooking instructions, recipe titles, and categories, that can be used to train machine learning models.

The primary goal of recipe classification is to make it easier for users to find alternative recipes based on their preferred genre [5]. The classification of food recipes into genres makes it

kaggle.com/datasets/nazmussakibrupol/3a2mext), and Human-Annotated Subset (https://www. kaggle.com/datasets/nazmussakibrupol/3a2m-dataset-human-annotated-subset). The relevant 3A2M project code for this study is publicly available from the GitHub repository (https:// github.com/nazmussakib009/3A2M).

**Funding:** The author(s) received no specific funding for this work.

**Competing interests:** The authors have declared that no competing interests exist.

easier for users to find recipes based on their preferences and tastes, as well as enhancing the search functionality of online recipe databases. Recipe genre classification allows the creation of recommendation and recipe generation systems that can make better suggestions to users based on their preferred cuisine or ingredients. However, classifying recipes into different genres can be subjective and based on personal opinions, which can lead to inconsistencies in the classification process. Classifying recipes into few genres can result in oversimplification, while classifying recipes into too many genres can make it difficult for users to find the recipe they are looking for [5–8]. Therefore, in this study, we have used a machine learning-based approach to classify a large recipe dataset, taking advantage of the ability of machine learning algorithms to handle large amounts of data and identify patterns and relationships in the data [8]. Methods used to classify food recipes include traditional machine learning, deep learning models and pre-trained language models.

This study has made several significant contributions to the applications of deep learning and natural language processing proposing a standardized framework for recipe genre classification.

- To begin with, we created a pipeline that generated a vast annotated dataset consisting of two million culinary recipes with extended named entities extracted from recipe descriptions. Additionally, we extended the Named Entity Recognition (NER) list to cover a wider range of named entities and their variations. By extending the named entity list of the 3A2M [9] dataset extracted from recipe descriptions (directions), we were able to address the issue of missing named entities like temperature of food processing, time of cooking and process of cooking or preserving. We named our resulting dataset the "Assorted, Archetypal, and Annotated Two Million Extended (3A2M+) Cooking Recipe Dataset".

- Secondly, the application of conventional machine learning, deep learning, and pre-trained language models was able to correctly classify recipe genres with an accuracy rate of 98.6%. This confirms the efficacy of pre-trained language models such as DistilBERT and RoBERTa in the classification of recipe genres.

- Finally, to classification, named entity recognition, and recipe generation, the dataset we have developed could also be useful in other recipe-related tasks such as recipe recommendation, ingredient substitution, dietary analysis, and recipe summarization. Therefore, the "Assorted, Archetypal, and Annotated Two Million Extended (3A2M+) Cooking Recipe Dataset" can serve as a valuable resource for various applications in the domain of culinary research and development. The 3A2M+ dataset will be available at this URL: https://www. kaggle.com/datasets/nazmussakibrupol/3a2mext.

The remainder of the paper is organized in the subsequent manner: Section 2 examines the necessary contextual information and previous research. The dataset description and the process of named entity extraction is discussed in section 3, while section 4 presents the proposed methodology. In section 5, the experimental setup is introduced, followed by the classification findings in section 6. Limitation of the research, conclusion and potential future work are discussed in section 7 and 8.

## 2 Literature review

One of the research challenges in this field is the automatic classification of recipes into different categories such as desserts, soups, or salads. Another challenge is the provision of personalized recipe recommendations based on user preferences and dietary restrictions. To address these issues, researchers have developed a variety of machine learning methods, including

transfer learning and active learning techniques. Moreover, the availability of large-scale recipe datasets has facilitated the development of advanced models capable of capturing complex relationships between ingredients, flavors, and cooking methods. This section covers a review of the literature on existing recipe datasets, recipe generation, machine learning techniques for recipe genre classification.

## 2.1 Recipe dataset

The availability of online recipe datasets has significantly transformed the way individuals discover and obtain recipes. These datasets, which are accessible through the internet, offer a vast collection of recipes, enabling people to conveniently search for recipes according to their preferences. Some of the commonly used online recipe datasets include Recipe1M+ [2], RecipeNLG [1], Food.com [10], RecipeDB [11], and Recipe5K [12]. With these datasets, people can easily find and access an extensive range of recipes, providing them with valuable information about ingredients, instructions, and even nutrition information. As such, these datasets have become a valuable resource for both home cooks and professional chefs alike.

Salvador et al. [13] proposed a method to learn cross-modal embeddings that can represent both food images and cooking recipes in a shared space, enabling the model to reason about the relationships between them. The authors used a deep neural network architecture to learn the cross-modal embeddings. The network consists of three components: a vision module that processes food images, a language module that processes cooking recipes, and a joint embedding module that combines the output of the vision and language modules into a shared space. The joint embedding module was trained to minimize the distance between the embeddings of corresponding food images and cooking recipes. To evaluate the effectiveness of their method, the authors conducted experiments on two datasets: Recipe1M and Food-101. The Recipe1M dataset contains over one million recipes with corresponding images, while the Food-101 dataset contains images of 101 food categories. The authors showed that their method outperformed several baseline methods in both the recipe-to-image and image-to-recipe retrieval tasks, indicating that their cross-modal embeddings were able to capture meaningful correlations between food images and cooking recipes.

Marin et al. [2] introduced a new dataset called Recipe1M+ for learning cross-modal embeddings for cooking recipes and food images. The authors argued that food is a highly visual and sensory experience, and that there is a need for models that can understand the relationship between food images and the corresponding recipes. The Recipe1M+ dataset [2] is created by combining two existing datasets, Recipe1M and the ECCV 2018 Food Image Recognition Challenge dataset. Recipe1M contains one million recipes with associated metadata, while the ECCV dataset contains 55,000 food images with associated labels. The authors used a combination of automatic and manual methods to align the recipes and images in the dataset. To demonstrate the usefulness of the Recipe1M+ dataset, the authors trained several models for cross-modal retrieval, including a recipe-to-image model and an image-to-recipe model. The authors showed that the models were able to retrieve relevant images and recipes based on a query, and that the quality of the retrieval was improved with the use of cross-modal embeddings learned from the dataset. The authors also provided a detailed analysis of the dataset, including statistics on the number of recipes, images, and unique ingredients in the dataset. They also provided examples of the types of relationships that can be learned from the dataset, such as the similarity between different types of pizza.

Generating natural language text that follows a semi-structured format, such as a cooking recipe, requires models that can understand the underlying structure and content of the text.

In order to develop and evaluate such models, large and diverse datasets are required. Bień et al. [1] presented RecipeNLG, a new dataset for semi-structured text generation in the domain of cooking recipes. The authors described their methodology for collecting and annotating the dataset. They first collected a large number of recipes from various sources, such as recipe websites and cookbooks. They then annotated the recipes by identifying and labeling the key components of the recipe, such as ingredients, cooking actions, and cooking times. The authors also provided statistics on the size and diversity of the dataset, including the number of unique ingredients, cooking actions, and recipe structures. The authors of this dataset conducted experiments on two tasks, namely recipe generation and recipe rewriting, to demonstrate the utility of the RecipeNLG dataset. To generate new recipes, they employed a neural language model, while for translating recipes from one language to another, they used a neural machine translation model. The authors were able to prove that the RecipeNLG dataset is valuable and can be utilized for training and assessing semi-structured text generation models for both tasks. They also compared the RecipeNLG dataset to other cooking recipe datasets and found that it is larger and more varied than previous datasets. They suggested that future studies can use the RecipeNLG dataset to enhance the quality and diversity of semi-structured text generation models.

Recipe5k [12] is a popular publicly available dataset of recipes that has been widely used in various research studies. The dataset contains around 5,000 recipes in English, each with an associated ingredient list and cooking instructions. Created by the Computer Vision Center at the Universitat Autònoma de Barcelona, Recipe5k has been used as a benchmark dataset to evaluate the performance of various recipe-related algorithms. One common use of Recipe5k is for recipe categorization, in which recipes are classified into different categories such as cuisine type, dietary restrictions, or meal type. It can be utilized to develop a recipe recommendation system that is based on the similarity of ingredients, demonstrating the effectiveness of the system in generating personalized recipe recommendations. Another important use of Recipe5k can be ingredient recognition, in which a system is trained to recognize ingredients from the text of the recipe. Overall, Recipe5k has become a widely used dataset in the field of recipe-related research, offering valuable resources and opportunities for researchers to evaluate and develop new algorithms and systems.

The 3A2M dataset [9] has been derived from the RecipeNLG dataset. While the RecipeNLG dataset contains recipe titles, directions, and Named Entity Recognition (NER) attributes without classification labels. So we have extended the ReceipeNLG dataset by adding genre as an important category. Three human experts classified 300K recipes into nine categories based on NER, with remaining 1900K recipes automatically classified using active learning and a query by committee approach. The dataset includes over two million recipes, each classified into one of the nine predefined genres. Preprocessing techniques such as unique word discovery, genre principle categorized word matching, and lowercase English letter conversion were used on the original unlabeled data using Natural Language Processing (NLP) techniques. The Fleiss Kappa score for the nine genres is around 0.56026.

We believe that 3A2M+ dataset introduced in this study is a valuable resource for those working on recipe classification and named entity recognition (NER) tasks, as well as other applications of recipe data. Its consistent format can help extract relevant information for classification and NER tasks, and the pre-processed annotations for several tasks can save time and effort in preparing the dataset. The flexibility of the dataset can also allow for exploration of different applications of recipe data beyond classification and NER. Overall, the 3A2M+ dataset can provide a diverse set of recipes to train and test models for various recipe-related tasks.

## 2.2 Recipe classification

The absence of openly available recipe datasets suitable for machine learning methods has obstructed progress in data-driven culinary research [14]. Although certain online recipe databases have made use of data-driven techniques to promote culinary research, there is a scarcity of a sizable annotated dataset of recipes categorized by genre that can help create dependable machine learning models and promote the development of this area of research.

Britto et al. [15] proposed a text analysis method to classify Brazilian Portuguese cooking recipes due to the need for personalized recipe recommendations and improved nutritional profiles. The authors manually categorized 1,080 recipes into seven categories and applied text mining techniques such as stemming, stop words removal, and TF-IDF to extract features. Machine learning algorithms including Naïve Bayes, Random Forest, and Support Vector Machines were used to classify the recipes. The proposed approach achieved an accuracy of up to 94%, outperforming similar studies.

On a later work, Britto et al. [16] introduced a multi-label classification method to identify food restrictions in recipes due to the difficulty people with allergies, intolerances, or dietary preferences face in finding suitable recipes. The authors manually labeled 1,080 Brazilian Portuguese recipes with food restrictions and used text mining techniques like stemming, stop words removal, and TF-IDF to extract features. Multi-label classification algorithms, such as Binary Relevance and Classifier Chains, were used to predict the food restriction labels. The proposed method achieved an F1-score of up to 0.93, surpassing baseline methods, and an ablation study was conducted to analyze the features' contribution to performance.

Jayaraman et al. [17] aimed to analyze classification models for cuisine prediction using machine learning due to the increasing interest in multiculturalism and personalized food recommendations. They used a dataset of 20,000 recipes from 20 cuisines and applied text mining techniques and machine learning algorithms such as Naïve Bayes, Decision Trees, and Support Vector Machines to classify the recipes. The proposed approach achieved up to 80% accuracy, outperforming baseline methods, and the authors compared the algorithms in terms of accuracy, precision, recall, and F1-score. To give an overview of recipe genre categorization, Table 1 provides a summary of previous studies on recipe classification.

Pre-trained language models, such as BERT (Bidirectional Encoder Representations from Transformers) [18] have shown great success in a wide range of natural language processing tasks. While CNN, GNN [19], and RNN [20] models have shown success in other domains, we opted to focus on BERT and its variants, including DistilBERT, due to their state-of-the-art performance in text-based tasks such as NER and genre classification. However, to the best of our knowledge, currently we have not found any study that utilized pre-trained language models for food recipe genre classification. One possible reason is the lack of large-scale datasets specifically designed for this task, which makes it difficult to train and fine-tune such models.

**Table 1. Summary of previous works on recipe classification.**

| Article | Description | Model Used | Number of Instances | Accuracy |
|---|---|---|---|---|
| [15] | Proposed a text analysis method to classify Brazilian Portuguese cooking recipes due to the need for personalized recipe recommendations and improved nutritional profiles. | Naïve Bayes, Random Forest, and Support Vector Machines | 1080 Brazilian recipes, 7 classes | 94% |
| [16] | Introduced a multi-label classification method to identify food restrictions in recipes due to the difficulty people with allergies, intolerances, or dietary preferences face in finding suitable recipes. | Text mining techniques like stemming, stop words removal, and TF-IDF | 1080 Brazilian recipes, Binary classification | 93% |
| [17] | Analyzed classification models for cuisine prediction using machine learning due to the increasing interest in multiculturalism and personalized food recommendations. | Naïve Bayes, Decision Trees, and Support Vector Machines | 20,000 recipes, 20 cuisines | 80% |

Additionally, recipe classification may require domain-specific knowledge and understanding of cooking terminology and techniques, which may not be effectively captured by pre-training on general text corpora. Moreover, traditional feature extraction techniques and machine learning algorithms have shown good performance in this task, which may lead to a preference for these methods. Therefore, in this work, we have focused on finding how large language models perform on recipe genre classification task.

## 3 3A2M+ dataset

This section provides a detailed description of the 3A2M+ dataset, construction process of an extended Named Entity Recognition (NER) list, compilation of dataset statistics, and comparison of the dataset with other existing datasets. 3A2M+ dataset incorporated all the data and features of 3A2M dataset and extended the named entity recognition list.

### 3.1 Data collection

Our data was sourced from the 3A2M dataset, which served as the base for our collection efforts. As we discovered some missing NER in this dataset, we extended the NER list by including previously absent ingredients, process names, temperature data, and cooking materials. As a result, we named the extended dataset 3A2M+ to reflect these changes. The 3A2M dataset [9] is derived from RecipeNLG, and contains a vast collection of 2,231,142 culinary recipes, making it the largest publicly available dataset of its kind. One of the limitations of RecipeNLG was the absence of specific genre categorization for the recipes, that is what the 3A2M dataset contains through the human annotation and active learning techniques [21]. Each recipe consists of a title, a list of ingredients with quantities, and step-by-step instructions. The recipe title provides an accurate description of the dish, while ingredient quantities are adjusted for serving sizes, and the corresponding measurement units are linked. The recipe instructions outline the various steps involved in preparing the dish, with the correct amount of each ingredient utilized. However, the structure of recipe titles in the 3A2M dataset [9] has some limitations. The authors did not find a consistent classification system, they identified a Named Entity Recognition (NER) list of ingredients, which is not comprehensive since the same ingredient often appears in multiple recipes. Expanding the NER list is one possible way to improve the 3A2M dataset, as the current list has a limited number of ingredients and may not recognize some. By increasing the size of the NER list, the dataset could more accurately identify ingredients and overcome one of its limitations.

### 3.2 Extended NER generation

Named Entity Recognition (NER) is a subtask of Natural Language Processing (NLP) that aims to identify named entities in text and classify them into predefined categories, such as people, organizations, locations, and products. Named Entity Recognition (NER) plays an essential role in 3A2M dataset [9], which is a collection of over 2 million recipes crawled from the web. The 3A2M dataset includes information such as recipe titles, ingredients, cooking instructions, nutritional information, and more. NER is used to extract entities from the recipe text, such as ingredients and cooking actions, and to classify them into different categories, such as food types, cooking methods, and tools.

The NER list of 3A2M dataset [9] may need enhancement because it currently only covers a limited set of named entities, such as ingredients, measurements, cooking actions, and tools. While these are important entities in the context of recipe generation, there may be other relevant entities that are not currently included in the list. For example, the dataset does not include information on the origin or cultural background of a dish, which could be useful in

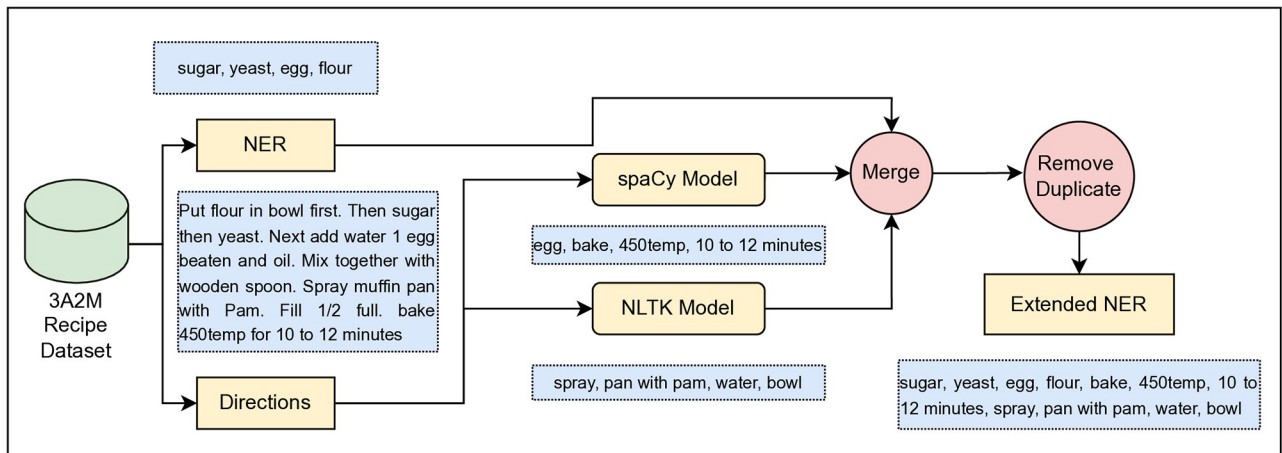

**Fig 1. Workflow of extended NER generation procedure.**

generating more personalized recipe recommendations for users. Additionally, the current NER list may not capture all possible variations of the named entities, which could lead to inaccurate or incomplete results in the recipe generation process. Therefore, enhancing the NER list to include a broader range of named entities and their variations could improve the overall quality and relevance of the generated recipes. With this motivation, in this study, we propose a pipeline to extend the NER list of 3A2M dataset [9] and include it in our 3A2M + dataset. The pipeline is depicted in Fig 1.

At first, we maintain the Named Entity Recognition (NER) list obtained from the 3A2M dataset. We then go through each recipe direction and pass it to both NLTK [22] and spaCy [23] toolkits, which produce two separate sets of named entities. Finally, we combine the original NER list with the newly generated NER list, creating a single set without any duplicates.

The Natural Language Toolkit (NLTK) performs named entity recognition (NER) using machine learning algorithms. Specifically, NLTK's NER module uses a maximum entropy classifier to identify entities within a text. The process starts with tokenizing the input text, i.e., splitting the text into individual words and punctuation marks. Next, the module labels each token with its corresponding part of speech (POS) tag using a POS tagger. The NER module then extracts features from each token, such as the word itself, its POS tag, and its context within the text. The maximum entropy classifier is then trained on a labeled dataset, such as the CoNLL 2003 dataset, which contains pre-labeled examples of named entities in text. The classifier uses these labeled examples to learn patterns in the features and to predict the label of new, unlabeled text. During the prediction phase, the classifier considers the features of each token in the text and predicts whether it belongs to one of the predefined named entity types, such as person, location, or organization. Finally, the module produces an output in which named entities are marked with their respective entity types. NLTK also provides the option to train the maximum entropy classifier on user-defined datasets, allowing for customization to specific domains or tasks.

"spaCy" toolkit uses a neural network architecture for named entity recognition (NER), where a pre-trained model is used to identify and classify entities. The procedure for named entity recognition in spaCy consists of several stages. First, the text is broken down into individual tokens or words and each is given its corresponding part of speech. Then, using a dependency parser, spaCy examines the relationships between the words in the sentence.

**Table 2. Named entities recognized from an example by NLTK and spaCy.**

| Model | Recipe Directions | New NER |
|---|---|---|
| spaCy | Put flour in bowl first. Then sugar, then yeast. Next, add water, 1 egg beaten, and oil. Mix together with wooden spoon. Spray muffin pan with Pam. Fill 1/2 full. Bake at 450˚ for 10 to 12 minutes. Pour rest of the mixture in a Tupperware container and refrigerate. Stir well each time you use. No rising necessary. | "first", "1 egg", "bake 450˚", "10 to 12 minutes", "Tupperware" |
| NLTK | Put flour in bowl first. Then sugar, then yeast. Next, add water, 1 egg beaten, and oil. Mix together with wooden spoon. Spray muffin pan with Pam. Fill 1/2 full. Bake at 450˚ for 10 to 12 minutes. Pour rest of the mixture in a Tupperware container and refrigerate. Stir well each time you use. No rising necessary. | "Spray", "Pam", "Pour", "Tupperware" |

spaCy extracts various characteristics of the text, such as its context, dependency relationships, and word structure, and employs these features to train the NER model. The model is based on a convolutional neural network (CNN) and a long short-term memory (LSTM) network, which are trained on a large corpus of text data. The model is capable of identifying different types of entities, such as people, organizations, locations, dates, and more, and can handle complex entity structures, such as nested and overlapping entities. The model is also able to learn from new examples and improve its accuracy over time. Once the named entities are recognized, spaCy performs further processing to enhance the accuracy of the results. This includes resolving named entity conflicts, merging overlapping entities, and refining entity boundaries. Table 2 shows the output of the NLTK and spaCy for a particular example recipe direction. Table 3 presents the final extended NER list obtained from our proposed pipeline.

## 3.3 Dataset properties

The Assorted, Archetypal and Annotated Two Million Extended dataset (3A2M+ dataset) is built on top of the 3A2M dataset [9] and we have incorporated as well as utilized all the data along with their respective features. In terms of recipe title, cooking step by step directions, ingredients, and recipe sources, all of the data are the same type. 3A2M+ dataset has in total five attributes: *title, directions, NER, Extended NER, genre*, and *label* among which the data of *title, directions, NER, genre, label* attributes are directly incorporated from 3A2M dataset.

In Table 4 number of the instances per genre are shown from the 3A2M+ dataset. Some sample instances from 3A2M+ dataset are displayed in the Table 5.

## 3.4 3A2M and 3A2M+ dataset comparison

The differences and advantages of the 3A2M+ dataset over the 3A2M dataset [9] are listed below:

- **Extended Named Entity Recognition**: The 3A2M dataset, developed from the RecipeNLG dataset, includes a limited list of named entities, which can result in incomplete or inaccurate extraction of entities from recipe texts. Notably, key NER attributes such as temperature,

**Table 3. Final NER list achieved from proposed extended NER pipeline.**

| Title | Direction | NER | Ext_NER |
|---|---|---|---|
| Pannu Kakku (Finnish Oven Pancake) | ["Preheat oven to 350 degrees.", "Melt butter in oven in a 9x13 pan; should be sizzling when you take it out.", "Meanwhile, mix other ingredients like hell—till very frothy. Pour batter into pan with melted butter.", "Bake 40 minutes. Eat immediately."] | "butter", "flour", "sugar", "eggs", "milk", "vanilla" | 'butter', '9x13', 'sugar', 'eggs', 'Bake 40 minutes', 'Melt', 'vanilla', '350 degrees', 'flour', 'milk' |

**Table 4. Total number of instances in the 3A2M+ dataset.**

| Genre ID | Genre Name | No of Instances in 3A2M+ | Human Annotated Data | Machine Annotated Data |
|---|---|---|---|---|
| 1 | Bakery | 160,712 | 28,481 | 132,231 |
| 2 | Drinks | 353,938 | 45,113 | 308,825 |
| 3 | NonVeg | 315,828 | 40,757 | 275,070 |
| 4 | Vegetables | 398,677 | 56,245 | 342,432 |
| 5 | Fast Food | 177,108 | 31,476 | 145,633 |
| 6 | Cereal | 340,495 | 45,677 | 294,818 |
| 7 | Meal | 53,257 | 7,009 | 46,248 |
| 8 | Sides | 338,497 | 37,210 | 301,287 |
| 9 | Fusion | 92,630 | 8,028 | 84,602 |
| **Total Data** | | **2,231,142** | **299,996** | **1,931,456** |

**Table 5. 3A2M+ dataset structure.**

| Title | Directions | NER | Extended NER | Genre | Label |
|---|---|---|---|---|---|
| No Bake Cheesecake | Mix cream cheese and sugar with electric mixer on medium speed until well blended. Gently stir in Cool Whip. Spoon into crust. Refrigerate 3 hours or overnight. | "cream cheese", "sugar", "graham cracker crust" | 'sugar', 'Mix', '3 hours', 'graham cracker crust', 'Spoon', 'cream cheese', 'Cool Whip', 'overnight' | Bakery | 1 |
| Lime Sherbet | Dissolve Jell-O in boiling water. Add sugar and lemon juice. When cool add milk. Freeze. If needed beat with mixer until smooth. | "lime Jell-O", "boiling water", "sugar", "lemons", "milk" | 'Add', 'sugar', 'lemons', 'milk', 'Dissolve Jell-O', 'Freeze', 'Dissolve', 'boiling water', 'lime Jell-O' | Drinks | 2 |
| Chicken Pot Pie | Cook chicken until no longer pink and cut up. Stir all ingredients together and put in pie shell (use deep dish pie pan). Cut slits in top crust. Bake at 375˚ for 40 minutes. | "cream of chicken soup", "cream of potato soup", "Veg-All vegetables", "milk", "pepper", "chicken breasts" | '375˚', 'Bake', '40 minutes', 'milk', 'Cook', 'Cut', 'cream of chicken soup', 'pepper', 'Veg-All vegetables', 'cream of potato soup', 'chicken breasts' | NonVeg | 3 |
| Granola Bars | Heat oven to 350˚. Mix all ingredients together except chocolate chips until moistened. Stir in chocolate chips. Press mixture evenly in ungreased 9x13-inch pan. Bake until center is set 15 to 17 minutes. | "Bisquick baking mix", "cooking oats", "brown sugar", "margarine", "egg", "semi-sweet chocolate chips", "raisins" | 'Stir', 'brown sugar', 'Bake', 'margarine', 'Bisquick baking mix', 'egg', 'semi-sweet chocolate chips', '13-inch', 'cooking oats', '9', '15 to 17 minutes', 'raisins' | Cereal | 6 |

cooking procedure, and quantity are absent in the original RecipeNLG dataset, although these are essential for accurate recipe categorization. The enhanced 3A2M+ dataset addresses this limitation by incorporating an extended list of named entities, covering more specific ingredients and cooking techniques, thereby improving the accuracy of NER tasks performed on the dataset.

- **Recipe Recommendation**: Since both the datasets are labeled with genre categories, machine learning models can be used for genre classification. This means that given a new recipe, a model can predict which genre category it belongs to based on the features of the recipe. This can have several benefits in recipe recommendation and personalization. For example, a user may have a preference for a specific genre of recipes, such as Italian or Mexican. By using a genre classification model trained on the 3A2M dataset [9], a recipe recommendation system can personalize its recommendations to the user's preferred genre. Additionally, a user may be looking for recipes for a specific occasion or meal type, such as a breakfast recipe or a recipe for a dinner party. By using the genre classification model, the recommendation system can narrow down the list of possible recipes to those that are most appropriate for the user's specific needs.

- **Availability**: Like 3A2M, the 3A2M+ dataset is also publicly available, making it more accessible for researchers and developers.

In summary, the 3A2M+ dataset provides significant enhancements that can provide more accurate and granular information about recipes, allowing for more targeted analysis and modeling based on specific aspects of recipes.

## 4 Methodology for recipe genre classification

In this study, we have used various traditional machine learning models, such as logistic regression, support vector machines, random forests, and naive Bayes, as well as a five-layer convolutional neural network to classify recipes into nine distinct genres. Additionally, we have employed two pre-trained language models, RoBERTa and DistilBERT, and we believe that we are the first to use transformer-based language models for classifying cooking recipe genres. In this chapter, we discuss our proposed methodology for cooking recipe genre classification on 3A2M+ dataset.

The proposed methodology depicted in Fig 2 for classifying cooking recipes into nine genres combines machine learning, transformer, and deep learning models. The input data, which

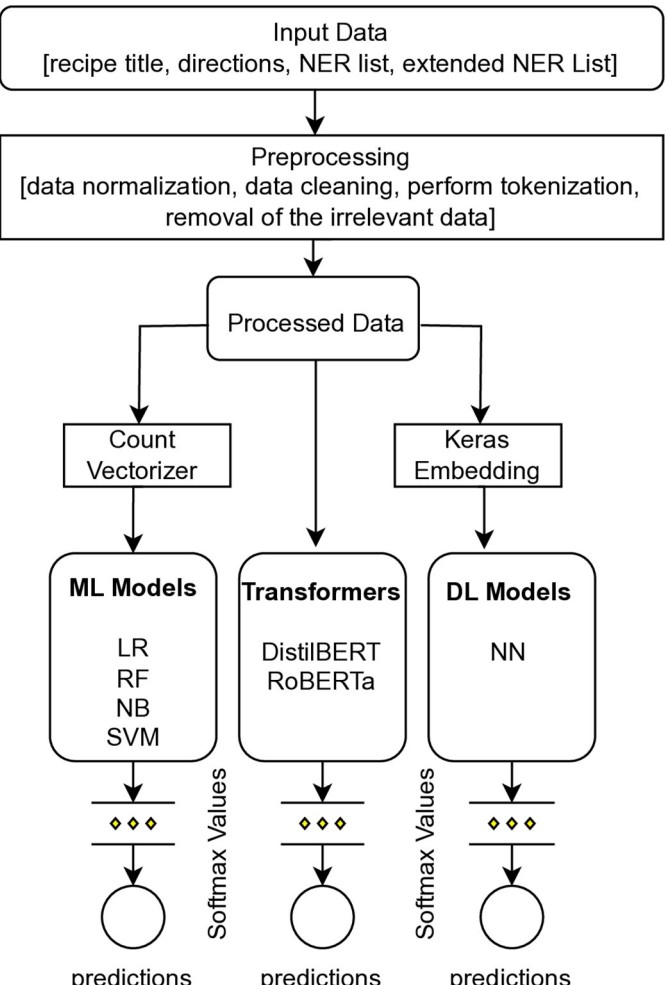

**Fig 2. Framework for recipe genre classification.**

can include recipe title, directions, NER list, and extended NER list is preprocessed and transformed using Count Vectorizer before being used to train and test different models. The final prediction is made by selecting the genre with the highest probability obtained through the softmax layer. This approach can be useful for recipe recommendation and menu planning.

To perform the classification task, the input data is preprocessed to ensure it is in a format that can be effectively analyzed. This step involves data normalization, cleaning, and tokenization to transform the raw data into processed data. The data is then divided into three subsets, training, validation, and testing sets, which are used to generate various features such as word embeddings and a document-term matrix.

After generating the features, machine learning models such as Logistic Regression, Random Forest, Naive Bayes, and Support Vector Machine are trained using the features obtained from the count vectorizer. Transformer models like RoBERTa and DistilBERT are also employed to generate input data representations based on pre-trained language models, which are then used as features for the softmax layer. A deep learning model is also used, which employs keras dense layer embeddings to train a Convolutional Neural Network (CNN) that recognizes patterns and associations in the input data. The final predictions are obtained by selecting the genre with the highest probability score from the softmax layer.

Overall, the workflow illustrated in Fig 2 allows for accurate classification of cooking recipes into nine different genres using a combination of machine learning, transformer, and deep learning models, which can be useful for various applications such as recipe recommendation and menu planning.

## 5 Experimental design

This section discusses the selection of baseline models, preprocessing techniques, and evaluation metrics for the baseline. The dataset was evaluated after construction to determine if machine learning models could distinguish between the classes. Two attention-based models were used, and a strong baseline performance was established. Analyzing the entire 2231K dataset took more than six months and involved over a thousand hours of work. Google Colab Pro Plus was used to tackle the computational challenge, but each set of results still took a long time to produce. After annotating the dataset, a machine learning model was created and applied to annotate the remaining 1900K rows of the dataset automatically. Traditional machine learning and language models were evaluated in the second phase. In the final phase, a new approach was taken using pre-trained models from a recipe dataset to contribute to NER. The combination of the old and new NER was a notable achievement. The collected data was analyzed using various machine learning algorithms.

### 5.1 Selection of baseline models

The baseline performance on 3A2M+ dataset is assessed using traditional machine learning models such as Logistic Regression [24], SVM [25], Naive Bias [26], and Random Forest [27], as well as a deep convolutional neural network (CNN) [28]. Additionally, we have utilized smaller, optimized pre-trained language models, including RoBERTa [29] and DistilBERT [30], which are built and trained over large BERT [18] architecture. This was done to consider the large size of the dataset and resource constraints. We have chosen RoBERTa and Distil-BERT models in this study for the following reasons:

i.  BERT-like models were chosen over CNN, RNN [20], and GNN [19] architectures for culinary text classification due to their ability to capture deeper contextual relationships in text through self-attention mechanisms. Unlike CNNs and RNNs, which process data

sequentially or in fixed-sized windows, BERT variants analyze entire sentences in parallel and consider both left and right contexts simultaneously. This bidirectional approach allows BERT-like models to understand nuances and dependencies between words more effectively, which is especially valuable in recipe texts where specific ingredient interactions, cooking techniques, and contextual meanings are crucial for accurate classification. Additionally, BERT's pre-training on vast amounts of text data makes it well-suited for domain-specific tasks like recipe genre classification [31].

While CNNs, RNNs, and GNNs are highly effective in structured or sequential data tasks, such as DNA and protein sequence analysis, they may face challenges in capturing the long-range dependencies and nuanced relationships characteristic of natural language. Recipe texts require models to capture both local and global information, given the variety of ingredient combinations and cooking methods they contain. BERT's self-attention mechanism provides a comprehensive view of these relationships, making it more suitable for this domain compared to traditional models like CNN, RNN, or GNN.

ii. Previous research has demonstrated that fine-tuning BERT models can produce remarkable results in tasks like text categorization and question-answering, due to the large amount of unlabeled data used in their pre-training through self-supervised learning. These BERT-based models have proven to be particularly effective in categorizing social media posts or comments. BERT-based models have demonstrated impeccable performance in the domain of categorizing social media posts or comments, for example, sentiment analysis of social media posts [32], political social media message categorization [33], rumor identification from tweets [34]. These models mitigate different limitations of previous state-of-the-art language models like ELMO [35] by adopting the transformer encoder instead of the recurrent neural network architecture.

iii. Implementing a recipe genre classification system for food recipe title, direction texts on devices with limited computing power can be challenging because of the high number of parameters in BERT (Base: 110 million), which increases the demands for computation and time during both training and inference. To mitigate this challenge, RoBERTa and DistilBERT can be used instead, as they offer comparable performance to BERT while having almost 40% fewer parameters and 60% less inference time, making it possible to run the system on edge devices.

## 5.2 Experiments

Cooking recipe genre classification is a challenging task due to the large variations in recipe styles and formats. In this paper, we perform four different experiments to explore the performance of various models on cooking recipe genre classification. Through these four experiments, we aim to develop an accurate and efficient genre classification model that can be utilized in recipe search engines, recommendation systems, and other food-related applications. Our experimental results shed light on the effectiveness of traditional machine learning models, deep learning models, and pre-trained language models in cooking recipe genre classification, and the contribution of different features in improving genre classification accuracy.

- In Experiment I, we aim to investigate the effectiveness of traditional machine learning models and a deep learning model on genre classification. We use Logistic Regression, Naive Bayes, Support Vector Machine, Random Forest, and Convolutional Neural Network (CNN) models, and several combinations of title, Named Entity Recognition (NER), and Extended NER features. Our objective is to explore the contribution of different features in

genre classification, specifically the title, title with NER, and title with Extended NER. The NER and Extended NER features consist of ingredients or process names. As the work is text-based, we utilize deep learning models and pre-trained language models to ensure that our genre classification creates a pipeline model to classify foods from any genre.

- In Experiment II, we aim to improve the performance of genre classification by using the DistilBERT classifier with several combinations of features. We explore the impact of title, NER, Extended NER, and Directions features on genre classification accuracy, with a random distribution of the recipe data.

- In Experiment III, we investigate the effect of equalizing the distribution of recipe data on genre classification accuracy. We use the DistilBERT classifier and several combinations of title, NER, Extended NER, and Directions features, but this time with an equalized distribution of the recipe data.

- In Experiment IV, we use the RoBERTa pre-trained language model with a single feature, Directions, to classify cooking recipe genres. Our objective is to determine the effectiveness of a pre-trained language model on genre classification and if equalizing the distribution of the recipe data would impact the accuracy of the classification.

## 5.3 Experimental settings and pre-processing

The effectiveness of BERT-based models in categorizing social media user comments is due to their pre-training on vast collections of text from various domains. As a result, the proposed dataset only needs minimal preprocessing to eliminate any unnecessary white spaces or tabs from the recipe title, NER, Extender NER, and Direction texts. Important variables are then defined and the Dataset class is created to specify how text is processed before being sent to the neural network. Data Loader is also defined, which is used to provide data to the neural network in appropriate batches for training and processing. The Dataset and Data Loader are both PyTorch elements that manage data preprocessing and transport to the neural network, with control options like *batch size* and *max_length*. The dataloaders are used in the training, testing and validation phases. The Training Dataset, consisting of 80% of the original data, is used for fine-tuning the model. The Validation Dataset is used to evaluate the model's performance and contains data that the model has not seen during training. The aim of optimizing classification performance is to fine-tune the pre-trained model weights to the specific task at hand, which involves the recipe title, NER, and directions texts, along with their annotated labels. As these models have been pre-trained using data from different sources, this is important. The experiment's training parameters are established first, and then the input is fine-tuned using a specific methodology. The recipe titles and directions are properly formatted before being fed into the pre-trained models for embedding. In order to represent the entire text input as a single vector, BERT-based models use tokenizers to divide the input sequence into word fragments or wholes. These token strings are used to identify related words and comprehend the context of the input sequence. Several types of specific token strings are used to indicate the task type, the start of the input sequence, the mask, and other factors [36].

- '[SEP]' refers to the end of one input sequence and the beginning of the following.

- '[PAD]' is employed to denote the required padding.

- '[UNK]' is an unidentified token.

- '[CLS]' refers to the classification task.

The classifiers require input sequences of the same length, which means that each title-NER combination formed for cross-embedding must have the same number of tokens after being converted into token strings. If a comment has less than 256 tokens, "[PAD]" tokens are added to achieve the 256-token limit. The maximum length for directions is 512 characters. During fine-tuning, some additional input data that was not included in DistilBERT and RoBERTa's pre-trained token vocabulary may arise. In these cases, the new input substring is replaced with "[UNK]". Finally, the token strings are converted into token IDs represented as integers to generate the final input vector for the models.

## 5.4 Hyper-parameter settings

The proposed dataset must be split into 3 parts to facilitate model evaluation and fine-tuning: training, validation, and testing. A 80% of the recipes from each class are randomly selected for the training set, while the remaining tweets are divided equally between the validation and testing sets. Base-uncased [37, 38] versions of the pre-trained models with a total of 768 hidden output states are used for fine-tuning. The Categorical Cross-Entropy loss is preferred over other loss functions for classification applications due to its superior performance [39]. The 'AdamW' optimizer [40] is chosen because it is efficient and works well with a fixed weight decay. The learning rate was set to $1 \times 10^{-5}$ and that 20% of the steps are classified as warm-up steps, the training phase would utilize the first 20% of the steps to increase the learning rate from 0 to $1 \times 10^{-5}$. Throughout the period of fine-tuning, the model weights are modified a total number of times, represented by steps. Both of these models were fine-tuned in a supervised manner for 10 epochs on the proposed dataset with a training batch size of 128 in order to predict the recipe genre from recipe NER, Extended NER, and Directions and achieved excellent performance on all nine classes.

## 5.5 Evaluation metrics

In predictive classification, evaluation metrics play a crucial role in determining a model's performance. However, using only metrics such as precision, accuracy, or recall may lead to incorrect conclusions, particularly in severely imbalanced datasets, where high accuracy can be achieved without making any meaningful predictions [41, 42]. In such cases, using multiple evaluation metrics, including recall, is necessary to provide a comprehensive assessment of the model's performance. Unlike accuracy or precision, recall considers false negatives, which can be a more crucial factor in highly unbalanced datasets. Therefore, using only a single metric can be misleading, and a combination of evaluation metrics is necessary for accurate assessment [43, 44].

This study uses ROC curve and AUC-ROC as evaluation metrics to determine how well the models can distinguish between different classes. The ROC curve calculates the True Positive Rate (TPR) and False Positive Rate (FPR) for a series of predictions at various thresholds made by the model. The TPR shows the number of positive class samples correctly classified by the classifier, while the FPR shows the number of negative class samples misclassified by the classifier [45]. This data may be used to evaluate the model's ability to distinguish across classes.

## 6 Experimental results

The baseline classification results on the 3A2M+ dataset is introduced in this section. Experiments details stated in Section 5.2 whose are mainly divided into two parts. The first part involves experiments utilizing traditional classifiers, specifically Experiment I. The second part encompasses experiments using language based model DistilBERT, divided into three experiments (Experiment II, III, IV).

**Table 6. Experiment-I configuration details.**

| Features | Machine Learning Models | Training Instances | Validation Instances | Testing Instances |
|---|---|---|---|---|
| 1. Title<br>2. Title with NER<br>3. Title with Extended NER | 1. Logistic Regression (LR)<br>2. Support Vector Machine (SVM)<br>3. Naive Bayes (NB)<br>4. Neural Network (NN)<br>5. Random Forest (RF) | 1100K data annotated by the machine. We separated the data into 11 files, each containing 100K instances due to the massive computational cost. | From the training instances, 10% of the instances are used for validation for each of the machine learning algorithms. | 300K human-annotated instances are used for testing. Here, 11 files were created with equal sizes. |

**Table 7. Accuracy of different machine learning models for different features.**

| Feature | LR Train | LR Test | SVM Train | SVM Test | NB Train | NB Test | CNN Train | CNN Test | RF Train | RF Test |
|---|---|---|---|---|---|---|---|---|---|---|
| Title | 99.95% | 96.13% | 99.03% | 98.11% | 86.67% | 80.17% | 99.72% | 97.94% | 99.46% | 75.35% |
| Title+NER | 91.49% | 74.79% | 89.94% | 81.11% | 59.23% | 60.65% | 99.63% | 95.30% | 93.71% | 56.84% |
| Title+Extended NER | 99.96% | 95.28% | 90.67% | 81.09% | 78.20% | 61.43% | 99.61% | 95.33% | 97.43% | 76.89% |

## 6.1 Experiment I—Classification performance result on traditional machine learning models

To overcome computational time and processing difficulties resulting from the environmental setup, the data is separated in multiple data blocks, which is employed in the experiments. To analyze the data, we utilized five traditional machine learning models. The dataset consists of 11000K machine-annotated data, divided into 11 data blocks for training and 300K human-annotated data was utilized. The training to testing ratio was mostly 80:20. The configuration details of the Experiment I is illustrated in the Table 6. In Table 7 performances of differnet traditional machine learning model is shown by experimenting on *Title*, *Title with NER* and *Title with Extended NER* features. Classwise ROC and Recall graphs are stated the performance analysis in the Figs 3 and 4 for the *Title with NER* and *Title with Extended NER* respectively. In addition, Figs 5 and 6 demonstrated the accuracy and loss function of the neural network simulation for the *Title with NER* and *Title with Extended NER* respectively.

The classification performance of traditional machine learning on 1100K machine data for training and 300K human annotated data for testing is summarized in Table 8. Among the 1100K training data, 10% of the data is used for validation purposes. In Table 8 demonstrated the best performance model for the each of the feature with the highest accuracy points for the training and testing sessions. We found that the "title" is the most contributing feature from the numerical results. The analysis and justification of the experimental results are discussed in the "Discussion of the Experiment" section, which provides a comprehensive overview of the findings and explains the implications of the results.

## 6.2 Experiment II—Classification performance result on DistilBERT model over random distributed data

In this part of the experiment DistilBERT models is used to analyze the data. 1100K machine-annotated data for training and 300K human-annotated data for testing is utilized. The data is maintained a split ratio 80:20 for training and testing, and the training data was further divided

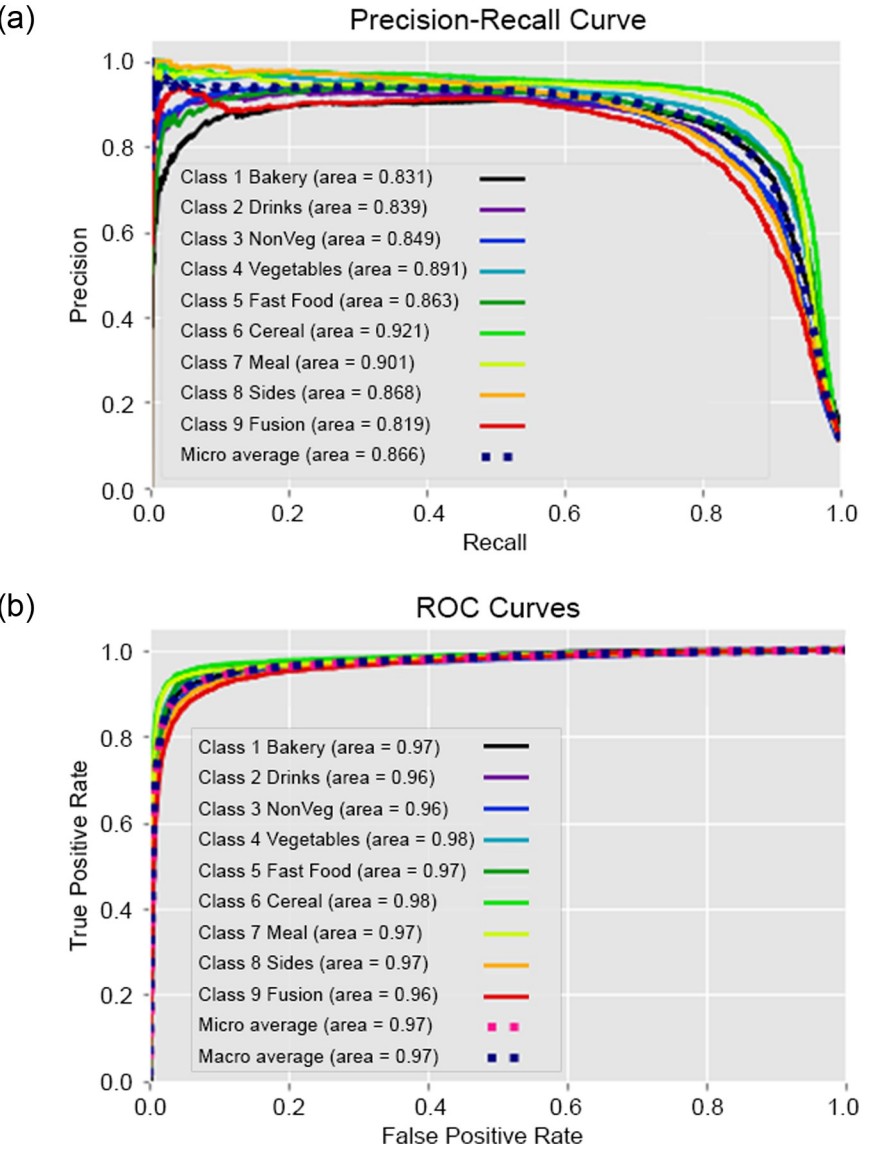

**Fig 3. Class-wise recall-ROC graph of Logistic Regression (LR) on title with NER feature.** (a) Recall Graph of Logistic Regression (b) ROC Graph of Logistic Regression.

90:10 for the validation issue. Experiment configuration details is presented Table 9. In this experiment 4 set of features were used. Begin with *Title* and followed by *Title with NER*, *Title with extended NER* and *Directions* feature from the 3A2M+ dataset.

Classification performance on language based model DistilBERT is summarized in the Table 10. It is important to analyze and discuss the results of experiments to draw meaningful conclusions and insights. In this table it picturize clearly that, *Title with NER* feature performed 98.31% accuracy in the training session, whereas *Title with Extended NER* feature performed 99.26% accuracy. In the validation of the data, which is showing the trend in the same direction, and in the testing phase, where human annotated data were utilized to check. There also discovered testing *Title with NER* feature testing accuracy 99.10% where *Title with Extended NER* performed accuracy 99.45% which is the obviously ahead of 0.35%. In the case

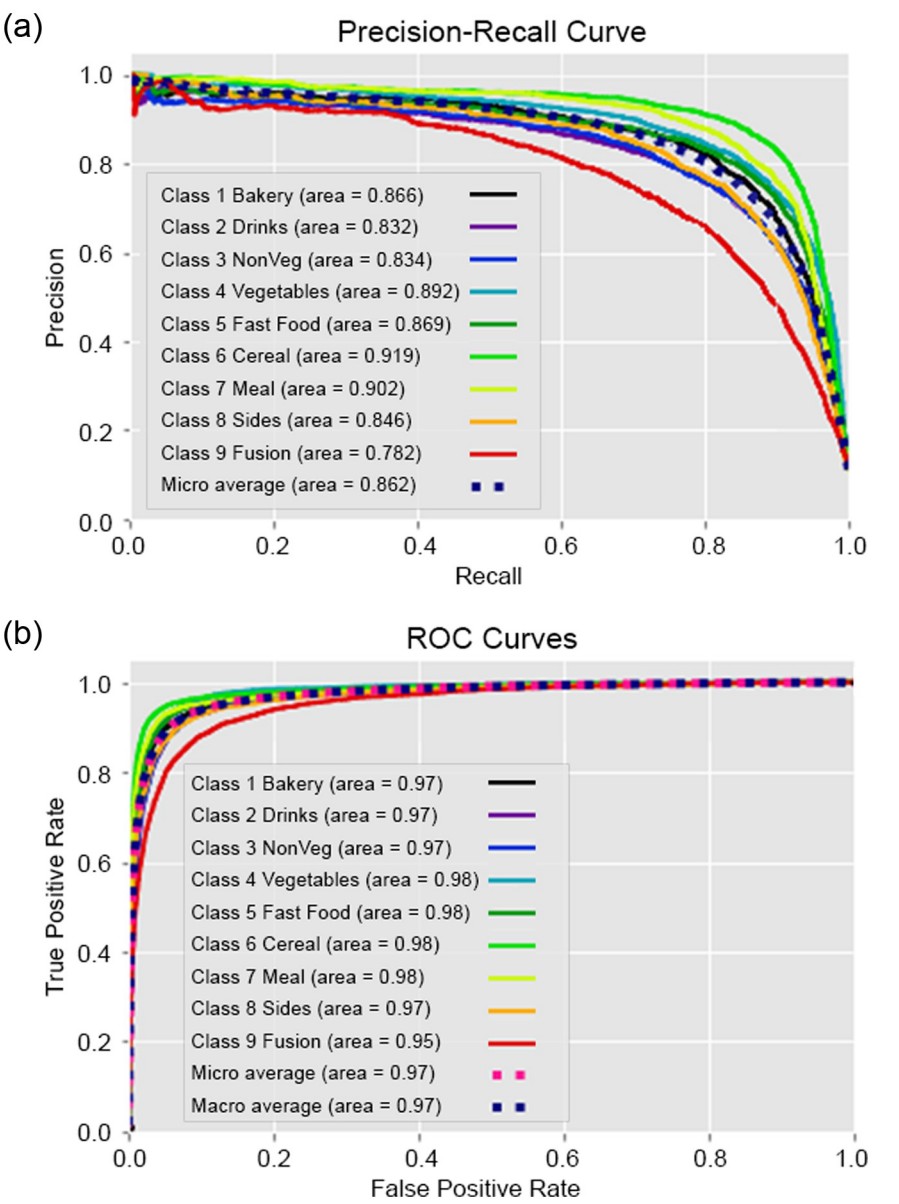

**Fig 4. Class-wise recall-ROC graph of Support Vector Machine (SVM) on title with extended NER feature.** (a) Recall Graph of SVM (b) ROC Graph of SVM.

of the experiments performed, the details of the analysis and evaluation of the results are discussed in the "Discussion of the Experiment" section.

In terms of percentage change, it is quite minor, but in terms of total dataset performance, it is a substantial shift and a major reflection of the study we have conducted. In the Fig 7 shows the differentiate the significance of the implementation of the *Title with NER* and *Title with Extended NER*. This study outcome is symbolized on that figure with some distinctions. The dataset using the DistilBERT model with a maximum length of 512, considering the system's maximum capacity. The maximum length of directions in the dataset is 2416 words. Here the data volume is huge so multiple times run performed and took the average value from the test runs.

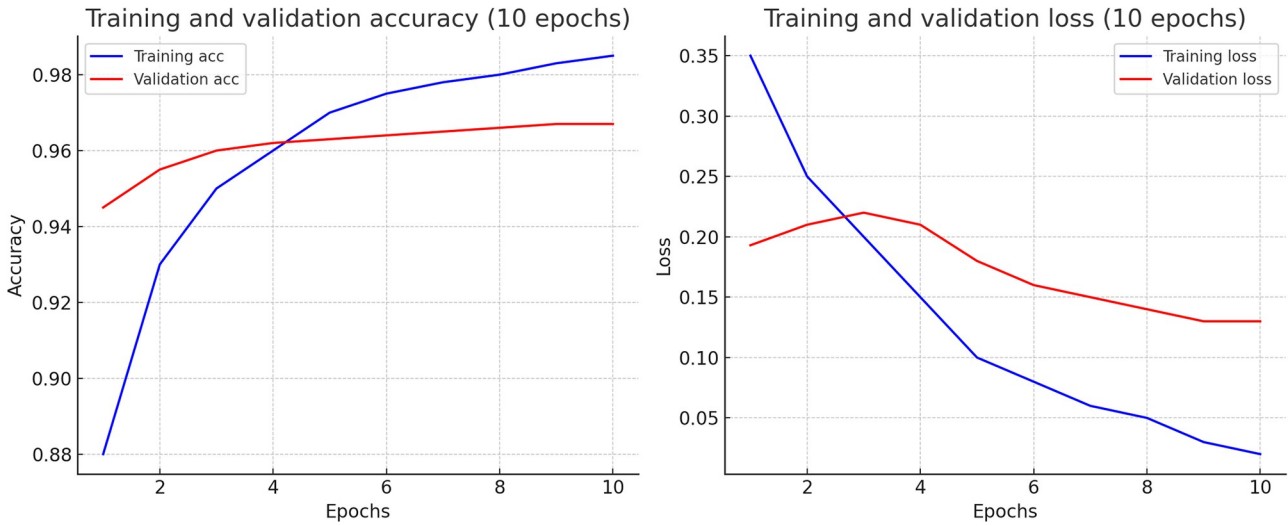

**Fig 5. Training and validation accuracy and loss graph on title with NER feature.**

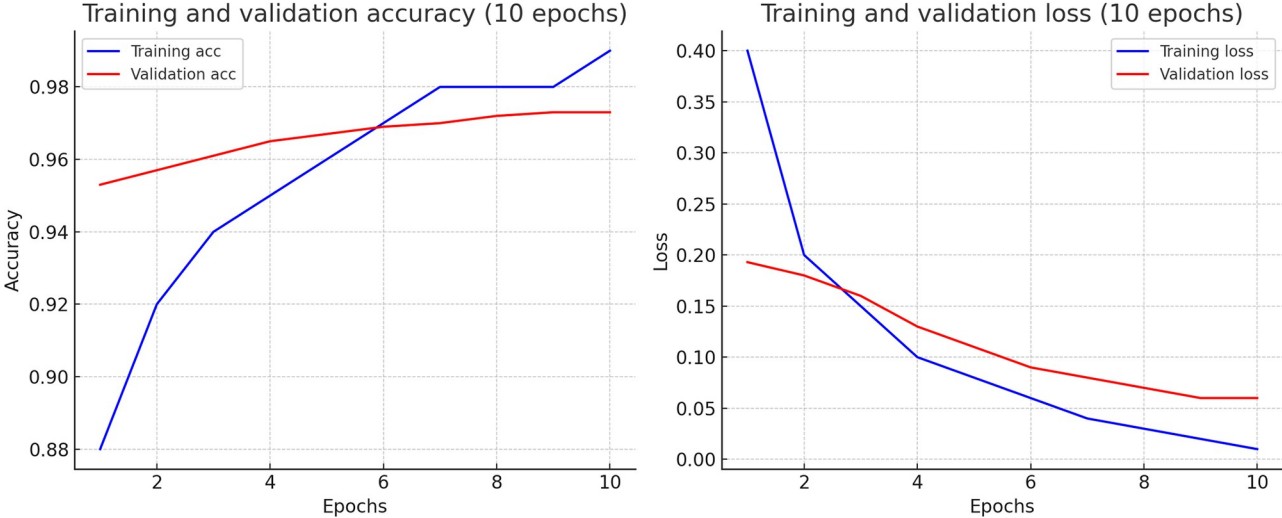

**Fig 6. Training and validation accuracy and loss graph on title with extended NER feature.**

## 6.3 Experiment III—Classification performance result on DistilBERT model over equalized data

Some ambiguity was noticed when using the directions feature while analyzing the whole dataset. Out of 1900K machine-annotated data, 46K instances was found in a specific genre. To

**Table 8. Dataset analysis summary on classical machine learning models.**

| Features | Training Performance | Testing Performance | Best Performance Model |
|---|---|---|---|
| Title | 99.03% | 98.11% | Support Vector Machine |
| Title+NER | 99.41% | 95.31% | Convolutional Neural Network |
| Title+Extended NER | 99.61% | 95.33% | Convolutional Neural Network |

**Table 9. Configuration details of Experiment II using DistilBERT model.**

| Title Feature | Data | Remarks |
|---|---|---|
| Training Dataset Size | 1,100,000 | Machine Annotated Data |
| Train Data | 1,000,000 | Machine Annotated Data |
| Validation Data | 100,000 | Machine Annotated Data |
| Test Data | 299,998 | Human Annotated Data |

**Table 10. Experiment II on dataset using DistilBERT model.**

| Features | Training Performance | Validation Performance | Testing Performance |
|---|---|---|---|
| Title | 97.89% | 98.17% | 99.14% |
| Title+NER | 98.31% | 97.99% | 98.86% |
| Title+Extended NER | 99.26% | 98.13% | 99.45% |
| Directions | 86.32% | 51.06% | 48.79% |

address this issue, this experiment created a configuration for training dataset by taking 46K of data from each of the 9 genres, resulting in a total of 414K data. The training data was split 90:10, with 10% used for validation. For testing, 27K data was used from the human-annotated data. This experiment is analyzed through four features *Title, NER, Extended NER, and Directions* by DistilBERT model over equalized data. For the first three features, a maximum length of 256 words was used, but for the Directions feature, which was too long to handle, used 512 words. To handle the large amount of data in the Directions genre, was divided the 414K data into 4 groups and processed each group with 103500 data and 10 epochs in a single fold operation. The experimental setup is illustrated in Table 11.

This work obtained the precision, recall, and F1-score for the classification of the *Title with NER* feature, separated by genre. In the Table 12 is illustrated the details of the experiment. Besides this, in the Table 13 is presented the details of the experiment of recall, and F1-score for the classification of the *Title with Extended NER* feature.

The experiment with an equalized data distribution, created a total intances of 414K. In this case, this study discovered that the *Title with NER* feature performed 98.99% in the training session, whereas the *Title with Extended NER* performed 99.08%. While the validation outcome was negative with very tiny accuracy, the testing accuracy for *Title with NER* feature 98.86% where *Title with Extended NER* performed accuracy 98.98% which is far above of 0.12%. Despite the fact that any range of data *Title with Extended NER* perform greater

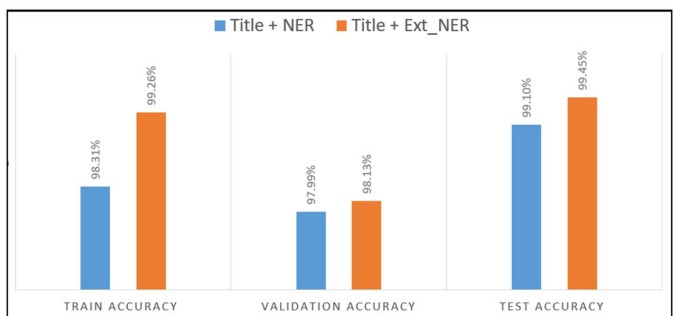

**Fig 7. Comparison of title with NER and extended NER.**

**Table 11. Configuration of Experiment III—Using DistilBERT over equally distributed data.**

| Title Feature | Data | Remarks |
|---|---|---|
| Training Dataset Size | 414,000 | Machine Annotated Data |
| Train Data | 103,500 | Machine Annotated Data |
| Validation Data | 10,350 | Machine Annotated Data |
| Test Data | 27,000 | Human Annotated Data |

accuracy, this is a significant addition. Table 14 and Fig 8 have shown the details of the difference between the *Title with NER feature* and *Title with Extended NER* feature experiments.

The results of the machine learning classification based on language, using 414K machine annotated data for training and 27K human annotated data for testing. 41K of the training data was used for validation purposes. Experiment results are summarized in Table 15. However, gaining a comprehensive understanding of the performance of the models and features used in the experiments requires analyzing and discussing the experimental results, which can be found in the "Discussion of the Experiment" section.

## 6.4 Experiment IV—Classification performance result on RoBERTa model over equalized data

The *direction* feature is evaluated with 1100K data for training and 300K data for testing and found that there was an under-fitting issue, with a large difference between the training and

**Table 12. Genre-wise precision, recall, and F1-score of title NER feature classification.**

| Genre ID | Genre Name | Precision | Recall | F1-Score |
|---|---|---|---|---|
| 1 | Bakery | 0.98 | 0.98 | 0.98 |
| 2 | Drinks | 0.98 | 0.98 | 0.98 |
| 3 | NonVeg | 0.99 | 0.99 | 0.99 |
| 4 | Vegetables | 0.99 | 1.00 | 0.99 |
| 5 | Fast Food | 0.99 | 0.99 | 0.99 |
| 6 | Cereal | 0.99 | 0.99 | 0.99 |
| 7 | Meal | 1.00 | 0.97 | 0.99 |
| 8 | Sides | 1.00 | 0.99 | 1.00 |
| 9 | Fusion | 0.98 | 0.99 | 0.98 |

**Table 13. Genre-wise precision, recall, and F1-score of title and extended NER feature classification.**

| Genre ID | Genre Name | Precision | Recall | F1-Score |
|---|---|---|---|---|
| 1 | Bakery | 0.97 | 0.99 | 0.98 |
| 2 | Drinks | 0.98 | 0.98 | 0.98 |
| 3 | NonVeg | 0.99 | 0.99 | 0.99 |
| 4 | Vegetables | 0.99 | 0.99 | 0.99 |
| 5 | Fast Food | 0.99 | 0.99 | 0.99 |
| 6 | Cereal | 1.00 | 0.99 | 0.99 |
| 7 | Meal | 1.00 | 0.98 | 0.99 |
| 8 | Sides | 1.00 | 0.99 | 1.00 |
| 9 | Fusion | 0.97 | 0.99 | 0.98 |

**Table 14. Comparative analysis of *Title with NER* and *Title with Extended NER* features with DistilBERT model over equalized distribution data.**

| Feature | (Title + NER) | (Title + Extended NER) | Remarks |
|---|---|---|---|
| Training Dataset Size | 416,000 | 416,000 | Machine Data |
| Feature | Title, NER | Title, Extended NER | Updates |
| Maxlen | 256 | 256 | Cover NER |
| Train Data | 374,400 | 374,400 | Machine Data |
| Validation Data | 41,600 | 41,600 | Machine Data |
| Test Dataset | 27,000 | 27,000 | Human Data |
| Embedding | Cross | Cross | Position |
| Train Accuracy | 98.99% | 99.08% | +0.09 |
| Validation Accuracy | 97.55% | 97.49% | -0.06 |
| Test Accuracy | 98.86% | 98.98% | +0.12 |
| Ratio | 90:10 | 90:10 | Increase |

testing results in Table 10. To address this, an equal distribution of the genre based data is taken and re-evaluated the results in the Table 15, which is also showed an even larger gap compared to the previous analysis. As a result, a decision is been taken to use another pre-trained RoBERTa model, for further analysis. The results of using RoBERTa, a BERT based model, on 416K data for training and 27K data for testing are shown in Table 16. However, the average result was not promising and consistent with the previous poor results from the Distil-BERT analysis. "Discussion of the Experiment" section provides an in-depth analysis and evaluation of the findings, allowing for a deeper understanding of the strengths and limitations of the models and features, as well as potential areas for future work and improvements.

## 6.5 Discussion on experiments

In the analysis, various features such as *Title, NER, Extended NER* and *Directions* were used. DistilBERT and RoBERTa models have been used to evaluate the performance of the *Directions* feature. The analysis shows that the performance of the models is improved with the increase in the data. However, the results of the RoBERTa model were poor compared to the DistilBERT model for the Direction feature.

- In the Experiment I, five traditional machine learning models were used to analyze the title and genre, obtaining 99% accuracy during training and 98% accuracy during testing.

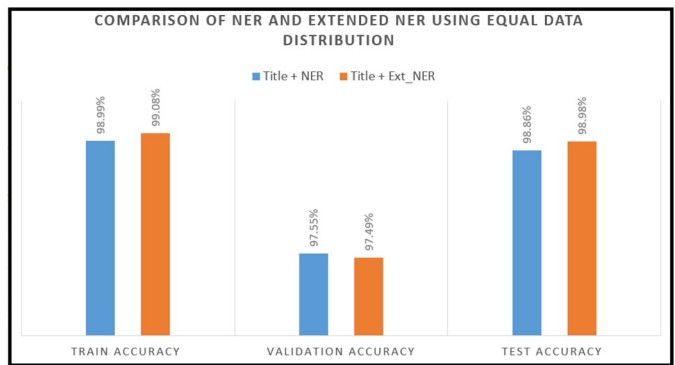

**Fig 8. Comparison of title with NER and extended NER over equalized data.**

**Table 15. Experiment III on dataset using DistilBERT model.**

| Features | Training Performance | Validation Performance | Testing Performance |
|---|---|---|---|
| Title | 96.83% | 97.06% | 97.21% |
| Title+NER | 98.99% | 97.55% | 98.86% |
| Title+Extended NER | 99.08% | 97.49% | 98.98% |
| Directions | 85.98% | 49.23% | 37.35% |

**Table 16. Direction feature classification by RoBERTa.**

| Results | Simulation-1 | Simulation-2 | Simulation-3 | Simulation-4 | Average |
|---|---|---|---|---|---|
| Training Accuracy | 49.56% | 11.34% | 34.27% | 47.23% | **35.60%** |
| Validation Accuracy | 49.22% | 10.46% | 33.12% | 46.12% | **34.73%** |
| Test Accuracy | 37.84% | 11.74% | 32.57% | 45.57% | **31.93%** |

Utilizing these five models to analyze the title, NER, and Extended NER, resulting in 95% accuracy during training and 90% accuracy during testing was presented. Our experimentation with various machine learning algorithms revealed that logistic regression, support vector machines (SVM), naive Bayes, and random forest were well-suited to the task of text classification. Additionally, a large training dataset of 100,000 examples enabled our model to generalize well to unseen data.

- The high accuracy achieved by our recipe genre classification model can be attributed to several factors, including the quality and nature of the dataset, the machine learning algorithms employed, and the features extracted from the text data. In natural language processing (NLP), the features extracted from the recipe titles play a crucial role in the performance of the model in Experiment I. Due to the short length and limited unique words in recipe titles, along with the similarity within titles of the same genre, our model can easily learn genre-specific patterns.

- In the Experiment II, with the use of a BERT based model DistilBERT, an achievement of a 99% accuracy in testing and 97% accuracy in training was resulted when analyzing the title. But in the Experiment III, with the use of same amount of instances in 9 genre we have achieved a narrow higher 97% accuracy in testing and close to 97% accuracy in the training over 416000 instances.

- In the Experiment II, by analyzing the title with NER, and title with Extended NER using the DistilBERT model, this experiment was obtained 98% and 99% accuracy in training respectively. In that same experiment we have found 98.86% and 99.45% accuracy in testing for the title with NER, and title with Extended NER features respectively.

- Relying solely on Named Entity Recognition (NER) or Extended NER for recipe classification proved to be insufficient as ingredients can be common to multiple recipes across different genres. Instead, combining the food title with the NER list resulted in more accurate genre classification results. Despite the short length of the recipe titles, they contain significant words that can specify the genre, whereas the similarity of NER lists or Extended NER lists are higher even though it can be a length of more than 47 words. By merging the title and NER or Extended NER, we created a larger word sequence in the Experiment II and III, which improved the performance of the title with NER or title with Extended NER, yielding

slightly lower accuracy than the title feature alone. Additionally, the extended NER list's inclusion of process type, temperature, and cooking style information further improved the model's classification performance.

- In the Experiment II, The Directions was selected for analysis using the BERT based model DistilBERT, with a maximum length of 512, in accordance with the system's maximum throughput. Test accuracy was less than 50% while train accuracy was around 85%, indicating a significant problem.

- Human annotated 27000 test data were used, equal in number from each of the nine genres, along with 103500 data, also equal in number from each genre were used in the Experiment III for the Directions feature. Training accuracy was 87% and testing accuracy was 47%.

- In the Experiment II analysis of 414,000 data with equal distribution was resulted a 0.35% improvement title with extended NER feature in accuracy compared to previous analysis of that only utilized title and NER, which remarked as a distinction of this research.

## 7 Limitations

A vast analysis is performed with the features in the 3A2M+ dataset. Fig 3 shows that the classes become more compact as the annotators' agreement approaches completion. The study is benefited from the expertise of experienced annotators from culinary and computer science domain, ongoing supervision, and communication between the annotators and subject-matter experts. There are some limitations of this work that is listed to improve future works.

- The annotated dataset was expected to have better quality if the annotators used the Extended NER.

- The Extended NER process was applied to the entire 3A2M dataset. During the initial creation of 3A2M, 1.9 million recipes were labeled through a machine learning process based on the original NER list alone. Reapplying this process with the Extended NER, which includes additional entities such as cooking methods, times, temperatures, ingredients, quantities, and tools, would make the 3A2M+ dataset even more robust. This enhancement would further strengthen the dataset's potential for accurate recipe categorization and analysis.

- Due to the time frame limitations of the Google Colab Pro+ hardware, this work had to limit the dimensions of word embedding in NER and Extended NER from the Directions.

## 8 Conclusion and future work

The present study presents a novel annotated dataset that incorporates an advanced feature for Named Entity Recognition (NER) of recipes. The dataset was created using robust reference sources and validated by food experts. The broader impact of this research could supports applications such as menu selection, recipe prediction, and customized menu designs by nutritionists and culinary experts to meet user needs.

The initial classification results for the dataset were obtained by optimizing pre-existing models, specifically DistilBERT. It's important to note that certain features of the dataset, such as directions and extended NER, were not considered during the annotation process. Enhancing the dataset's imbalance in terms of class distribution, incorporating more training examples, cleaning the data prior to training through pre-processing, and using more robust pre-trained models could all contribute to a more precise classification result.

Individuals can choose items from their kitchen or store and come up with meal names or categories that can be made using those ingredients. Some recipes might fit into multiple categories due to similarities in ingredients, however, they have been annotated based on expert opinions. The normalization of ingredient lists can help to address the issue of the same ingredient appearing in different forms. Adding additional metadata, such as cuisine type, meal type, or level of difficulty, could also improve genre categorization and enable the development of more specialized models. With its large size and categorization by genre, the medical field, particularly those dealing with food nutrition, can suggest a range of meals from the collection.

## Author Contributions

**Conceptualization:** Nazmus Sakib, Hasan Mahmud.

**Data curation:** Nazmus Sakib, G. M. Shahariar, Md. Mohsinul Kabir.

**Formal analysis:** Nazmus Sakib, G. M. Shahariar.

**Investigation:** Md. Mohsinul Kabir.

**Methodology:** Nazmus Sakib, Md. Kamrul Hasan, Hasan Mahmud.

**Supervision:** Md. Kamrul Hasan, Hasan Mahmud.

**Validation:** Hasan Mahmud.

**Writing – original draft:** Nazmus Sakib, G. M. Shahariar.

**Writing – review & editing:** Md. Mohsinul Kabir, Md. Kamrul Hasan, Hasan Mahmud.

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
