## [Decision Letter · Decision Letter 0]

18 Sep 2024

PONE-D-24-34319Towards Automated Recipe Genre Classification using Semi-Supervised LearningPLOS ONE

Dear Dr. Sakib,

Thank you for submitting your manuscript to PLOS ONE. After careful consideration, we feel that it has merit but does not fully meet PLOS ONE’s publication criteria as it currently stands. Therefore, we invite you to submit a revised version of the manuscript that addresses the points raised during the review process.

We look forward to receiving your revised manuscript.

Kind regards,

Zeheng Wang

Academic Editor

PLOS ONE

2. During your revisions, please note that a simple title correction is required: Springer Nature 2021 LATEX template has to be removed. Please ensure this is updated in the manuscript file and the online submission information.

4. Please update your submission to use the PLOS LaTeX template. The template and more information on our requirements for LaTeX submissions can be found at http://journals.plos.org/plosone/s/latex.

5. Please provide a complete Data Availability Statement in the submission form, ensuring you include all necessary access information or a reason for why you are unable to make your data freely accessible. If your research concerns only data provided within your submission, please write "All data are in the manuscript and/or supporting information files" as your Data Availability Statement.

6. Please provide a complete Data Availability Statement in the submission form, ensuring you include all necessary access information or a reason for why you are unable to make your data freely accessible. If your research concerns only data provided within your submission, please write "All data are in the manuscript and/or supporting information files" as your Data Availability Statement.

7. PLOS requires an ORCID iD for the corresponding author in Editorial Manager on papers submitted after December 6th, 2016. Please ensure that you have an ORCID iD and that it is validated in Editorial Manager. To do this, go to ‘Update my Information’ (in the upper left-hand corner of the main menu), and click on the Fetch/Validate link next to the ORCID field. This will take you to the ORCID site and allow you to create a new iD or authenticate a pre-existing iD in Editorial Manager.

Additional Editor Comments (if provided):

Reviewers' comments:

Reviewer's Responses to Questions

**Comments to the Author**

1. Is the manuscript technically sound, and do the data support the conclusions?

Reviewer #1: Partly

Reviewer #2: Partly

2. Has the statistical analysis been performed appropriately and rigorously? 

Reviewer #1: Yes

Reviewer #2: Yes

3. Have the authors made all data underlying the findings in their manuscript fully available?

Reviewer #1: No

Reviewer #2: No

4. Is the manuscript presented in an intelligible fashion and written in standard English?

Reviewer #1: Yes

Reviewer #2: Yes

5. Review Comments to the Author

Reviewer #1: The authors extend the 3A2M Cooking Recipe dataset to 3A2M+ by first creating a larger named entity list and then classifying recipes into genres. They combine the existing NER list with those found with a maximum entropy classifier and a neural network classifier. They classify genres based on the extended NER list using a variety of machine learning methods and pre-trained language models. This demonstrated the extended NER allowed for more accurate genre labeling, although this is unsurprising since it includes the original NER list.

Assuming the above is true, overall, the methodology, results, and specific conclusions appear consistent. The results need extra details and proper presentation, in particular all the results figures. Furthermore, the manuscript requires significant revisions for clarity as it contradicts itself and is ambiguous at several points.

My specific comments:

The authors need to be more explicit about what is previously published and what is novel, especially with regards to the 3A2M dataset. In particular, these two quotations seem to contradict each other and require more clarity:

Page 5 Paragraph 3:

“The 3A2M dataset [9] is based on the RecipeNLG dataset and incorporates all

of its data and features. The data attributes include title, directions, NER and genre

as labels”

Page 7 Paragraph 2:

“The 3A2M dataset [9] contains a vast collection of 2,231,142

culinary recipes, making it the largest publicly available dataset of its kind. One lim-

itation of the dataset is that it lacks specific genre categorization for the recipes.”

What is the problem with the 3A2M genre labels?

Also, where do the 3A2M+ dataset genre labels come from? Are they inherited from 3A2M? How accurate were they originally in 3A2M? If your models are more consistent with the human labelled dataset, did you update the 3A2M+ genre labels?

Furthermore, the original 3A2M dataset had human labelled data and machine labelled data. For all experiments, the authors use machine labelled data to train their network and test against human labelled data. Is there a limitation by the fact machine learning models are trained on data generated by previous machine learning models, rather than just the human labelled dataset?

Since the extended NER list includes the original NER list, the authors should state how the original NER list was created. This will allow the method to be extended to other databases or other genre labels.

Page 18, Figure 3 and 4:

The labels in the legend are too long. You can put “Precision-recall curves” in the figure caption and expand on it for clarity and simply label each curve by number. If I understand correctly, the 9 classes are the recipe genres, so it may be worth explicitly writing those in the legend. Furthermore, these figures are not discussed in depth in the text. What effect do they have on your results?

Page 19, Figure 5 and 6:

Axis labels are missing. I assume Accuracy against Epoch? If that is the case, I don’t understand why the validation accuracy is so high initially before training has begun, and plateaus while training. I also don’t understand why the validation loss gets worse while training either. Is this because you're training on machine labeled data but testing on human labeled data? It’s also unclear exactly which model this is for. Furthermore, you said there were only 10 epochs of training, but Figure 5 goes to 20 on the x-axis. These figures need clarification in the main text or caption.

Page 25 Limitations, second point:

“Domain experts resolved tie situations. If the study was to redo the Extended

NER process for the entire 1900K dataset, it would make the dataset more

robust.”

Does this mean the Extended NER process was not performed for the full 3A2M dataset? Or is this about not generating genre labels based on Extended NER? The sentence needs to be clarified or expanded.

The motivation and conclusion for genres in recipes says it can be used for identifying diet restrictions or the origin of dishes, however the specific genres used don’t seem designed for these and there's no connection to these uses. I understand this is a limitation of the original 3A2M dataset. It could be possible to connect the results to the current conclusion by, for example, commenting on the recall/ROC of the vegetarian class and the implications for people on vegetarian diets. Otherwise, proposing a way to extend the dataset to different genre labels, which would make the extended NER list look more useful. Otherwise, the conclusion should be written closer to the specific uses of the current genre labels, e.g. recipe prediction, menu selection and explicitly mention the limitations.

Some typographical errors I noticed:

Page 23 Section 6.5

“the Directionsfeature.” missing space

Page 24

“In the Experiment II, by analyzing the title with NER, and title with Extended

NER using the DistilBERT model, this experiment was obtained 98% and

99y% accuracy in training respectively” 99% has extra y

Data availability – authors selected no. However, they have released the database. For full reproducibility, the human labelled subset of the database should be identified or separately released. If this is already the case, it should be mentioned.

Reviewer #2: In response to the lack of sufficient labeling data, it can be challenging to categorize raw recipes found online into appropriate food types. The authors constructed a dataset called the “Classified, Prototyped, and Annotated 2 Million Extended (3A2M+) Cooking Recipes Dataset” and tested a variety of classification methods and tasks on this dataset. The results of this paper are fulfilling, but I have a few concerns.

1. The authors claim limited access in Data Availability, but as far as I know, building the dataset was the main work of this paper. And this dataset was collected from public data, it is the process and methodology of building it that is innovative and needs to be described further in the paper.

2. I think it is a jump to go from a traditional machine learning model and extend it directly to a pre-trained BERT model. The authors could have considered including experiments with CNN, GNN[1], RNN[2], and other models. Or explain the reasons why these methods were not chosen. These methods perform well in both DNA and protein tasks.

[1]Cao B, Wang B, Zhang Q. GCNSA: DNA storage encoding with a graph convolutional network and self-attention[J]. Iscience, 2023, 26(3).

[2]Li X, Han P, Chen W, et al. MARPPI: boosting prediction of protein–protein interactions with multi-scale architecture residual network[J]. Briefings in Bioinformatics, 2023, 24(1): bbac524.

6. PLOS authors have the option to publish the peer review history of their article (what does this mean?). If published, this will include your full peer review and any attached files.

Reviewer #1: No

Reviewer #2: No

---

## [Author Response · Author response to Decision Letter 0]

7 Nov 2024

Thank you to the editor and reviewers for their valuable feedback.

Editor: Thank you for your detailed instructions. We have adhered to the PLOS ONE LaTeX template, included author name formatting as required, provided the data availability link, and added links to the code repository within the manuscript. ORCID information is also included in Editorial Manager.

Reviewer 1: Thank you for the constructive comments. We have addressed each point in detail in the response document (Response to Reviewers) and reflected all changes in the manuscript. Typographical errors were corrected, and an image was replaced due to a file name error—our apologies for this oversight. The limitations and conclusion sections were revised following your guidance.

Reviewer 2: Thank you for the valuable comments. We clarified our model selection process in detail and have elaborated in the response document (Response to Reviewers). We have also cited the recommended paper as guided.

---

## [Decision Letter · Decision Letter 1]

17 Dec 2024

PONE-D-24-34319R1Towards Automated Recipe Genre Classification using Semi-Supervised LearningPLOS ONE

Dear Dr. Sakib,

Thank you for submitting your manuscript to PLOS ONE. After careful consideration, we feel that it has merit but does not fully meet PLOS ONE’s publication criteria as it currently stands. Therefore, we invite you to submit a revised version of the manuscript that addresses the points raised during the review process.

We look forward to receiving your revised manuscript.

Kind regards,

Zeheng Wang

Academic Editor

PLOS ONE

**Journal Requirements:**

**Additional Editor Comments:**

Please double-check to ensure that ALL datasets and codes used in this work can be freely accessed by other researchers for reproduction upon acceptance.

Reviewers' comments:

Reviewer's Responses to Questions

**Comments to the Author**

1. If the authors have adequately addressed your comments raised in a previous round of review and you feel that this manuscript is now acceptable for publication, you may indicate that here to bypass the “Comments to the Author” section, enter your conflict of interest statement in the “Confidential to Editor” section, and submit your "Accept" recommendation.

Reviewer #2: All comments have been addressed

2. Is the manuscript technically sound, and do the data support the conclusions?

Reviewer #2: Yes

3. Has the statistical analysis been performed appropriately and rigorously? 

Reviewer #2: N/A

4. Have the authors made all data underlying the findings in their manuscript fully available?

Reviewer #2: Yes

5. Is the manuscript presented in an intelligible fashion and written in standard English?

Reviewer #2: Yes

6. Review Comments to the Author

**Reviewer #2: **All comments have been address. I think this paper suit to publish in PLOS One in current Version. Thanks for Author's contribution.

7. PLOS authors have the option to publish the peer review history of their article (what does this mean?). If published, this will include your full peer review and any attached files.

Reviewer #2: No

---

## [Author Response · Author response to Decision Letter 1]

1 Jan 2025

This submission represents the second revision of our manuscript, and we have made comprehensive updates to address all the suggestions and comments provided by Reviewers and Editor. The reviewers have acknowledged that all raised issues have been adequately resolved in first revision. Specifically, we have ensured that all datasets and codes used in our study are publicly available, with verified and functional links included in the manuscript. Additionally, the references have been carefully reviewed and cross-checked to confirm that they are accurate and accessible in the public domain. Furthermore, we have revised the image formatting in accordance with the PACE guidelines, ensuring compliance with the journal's standards.

---

## [Editor Report · Decision Letter 2]

3 Jan 2025

Towards Automated Recipe Genre Classification using Semi-Supervised Learning

PONE-D-24-34319R2

Dear Dr. Sakib,

We’re pleased to inform you that your manuscript has been judged scientifically suitable for publication and will be formally accepted for publication once it meets all outstanding technical requirements.

Kind regards,

Zeheng Wang

Academic Editor

PLOS ONE
---

## [Editor Report · Acceptance letter]

16 Jan 2025

PONE-D-24-34319R2 

PLOS ONE

Dear Dr. Sakib, 

I'm pleased to inform you that your manuscript has been deemed suitable for publication in PLOS ONE. Congratulations! Your manuscript is now being handed over to our production team.

Kind regards, 

on behalf of

Dr. Zeheng Wang 

Academic Editor

PLOS ONE